# Target Values for 25-Hydroxy and 1,25-Dihydroxy Vitamin D Based on Their Associations with Inflammation and Calcium-Phosphate Metabolism

**DOI:** 10.3390/nu16162679

**Published:** 2024-08-13

**Authors:** Xitong Li, Yvonne Liu, Xin Chen, Christoph Reichetzeder, Saban Elitok, Bernhard K. Krämer, Berthold Hocher

**Affiliations:** 1Fifth Department of Medicine (Nephrology/Endocrinology/Rheumatology/Pneumology), University Medical Center Mannheim, University of Heidelberg, 69120 Mannheim, Germany; lixitong0729@gmail.com (X.L.); yvonne.liuye.2000@gmail.com (Y.L.); xin.chen@charite.de (X.C.); bernhard.kraemer@umm.de (B.K.K.); 2Department of Nephrology, Charité Universitätsmedizin Berlin, 10117 Berlin, Germany; 3Institute for Clinical Research and Systems Medicine, Health and Medical University, 14467 Potsdam, Germany; christoph.reichetzeder@hmu-potsdam.de (C.R.); saban.elitok@klinikumevb.de (S.E.); 4Department of Nephrology and Endocrinology, Klinikum Ernst von Bergmann, 14467 Potsdam, Germany; 5Reproductive and Genetic Hospital of CITIC-Xiangya, Changsha 410008, China; 6School of Medicine, Central South University, Changsha 410078, China

**Keywords:** 1,25-dihydroxy vitamin D, 25-hydroxy vitamin D, phosphate, iPTH, calcium, CRP, white blood cell count

## Abstract

Target values for 25-hydroxy vitamin D and 1,25(OH)_2_D or 1,25-dihydroxy vitamin D remain a topic of debate among clinicians. We analysed data collected from December 2012 to April 2020 from two cohorts. Cohort A, comprising 455,062 subjects, was used to investigate the relationship between inflammatory indicators (white blood cell [WBC] count and C-reactive protein [CRP]) and 25(OH)D/1,25(OH)_2_D. Cohort B, including 47,778 subjects, was used to investigate the connection between 25(OH)D/1,25(OH)_2_D and mineral metabolism markers (phosphate, calcium, and intact parathyroid hormone [iPTH]). Quadratic models fit best for all tested correlations, revealing U-shaped relationships between inflammatory indicators and 25(OH)D and 1,25(OH)_2_D. Minimal CRP and WBC counts were observed at 1,25(OH)_2_D levels of 60 pg/mL and at 25(OH)D levels of 32 ng/mL, as well as of 42 ng/mL, respectively. iPTH correlated inversely with both 1,25(OH)_2_D and 25(OH)D, while phosphate as well as calcium levels positively correlated with both vitamin D forms. Calcium-phosphate product increased sharply when 25(OH)D was more than 50 ng/mL, indicating a possible risk for vascular calcification. Multiple regression analyses confirmed that these correlations were independent of confounders. This study suggests target values for 25(OH)D between 30–50 ng/mL and for 1,25(OH)_2_D between 50–70 pg/mL, based particularly on their associations with inflammation but also with mineral metabolism markers. These findings contribute to the ongoing discussion around ideal levels of vitamin D but require support from independent studies with data on clinical endpoints.

## 1. Introduction

Vitamin D, a nutrient of growing importance for human health, exists predominantly as two metabolic derivatives: 1,25-dihydroxy vitamin D (1,25(OH)_2_D) and 25-hydroxy vitamin D (25(OH)D). The primary vitamin D reservoir form is 25(OH)D and is the most frequently assessed form of vitamin D in clinical examinations as a sign of the vitamin D level in the body. However, a highly bioactive form of vitamin D is 1,25(OH)_2_D. Inflammatory reactions and the metabolism of calcium and phosphate are two important processes that vitamin D influences. Within the metabolism of calcium and phosphate, 25(OH)D and 1,25(OH)_2_D facilitate the process of absorbing calcium and phosphate within the colon by augmenting the synthesis and efficacy of calcium ion channels and phosphate transport proteins [1,2,3]. Simultaneously, vitamin D modulates blood proportions of phosphate and calcium by regulating their reabsorption in the kidneys, thus maintaining a physiological balance of these essential minerals in the body [4,5,6,7].

Vitamin D exhibits a complex relationship with inflammation, characterized by bidirectional interactions. It suppresses the generation of inflammatory mediators while enhancing the production of anti-inflammatory cytokines [8,9], thereby modulating the immune system’s response and mitigating the extent and duration of inflammation. Furthermore, vitamin D controls how well immune cells operate and influences the inflammatory cascade [10,11]. Conversely, the inflammatory milieu can impact the effectiveness and metabolism of vitamin D. Inflammatory conditions disrupt the hepatic metabolism of vitamin D, resulting in diminished activity levels [12]. Specifically, white blood cells (WBCs) and C-reactive protein (CRP), serving as pivotal indicators of the immunological system and inflammation activity, are frequently utilized to investigate the relationship between vitamin D and inflammatory responses. A significant amount of research has demonstrated an inverse relationship between inflammatory indicators and 25(OH)D concentrations, including WBC and CRP [13,14], with a minor fraction revealing no correlation [15,16]. For 1,25(OH)_2_D, however, research in this area is limited.

Moreover, vitamin D is important for cardiovascular health [17] and neurological function [18]. Therefore, it is essential for human health to maintain the right level of vitamin D. Throughout the last 20 years, numerous studies have tried to find the lower and upper thresholds for vitamin D (see Table 1). Academic attention primarily centers around 25(OH)D, the content of which should not exceed 100 ng/mL in the blood—some authorities propose limits up to 300 ng/mL—while its lower boundary typically spans from 10 ng/mL to 30 ng/mL (see Table 1).

In this study, we analyzed data from a general German population to identify blood biomarkers associated with vitamin D, focusing on inflammatory responses and the metabolism of calcium and phosphate. Our goal is to ascertain the connection between vitamin D and parameters of the inflammatory response and calcium and phosphorus metabolism, as well as to determine reference target values for 25(OH)D and 1,25(OH)_2_D.

## 2. Materials and Methods

### 2.1. Study Population

This is a retrospective cohort study. Data were collected at the Institute for Medical Diagnostics Berlin, Germany, and we analyzed the data in two cohorts:

In cohort A, we explored the link between inflammation and 1,25(OH)_2_D and 25(OH)D. Patients from the database were included if they received 1,25(OH)_2_D and 25(OH)D measurements, and inflammatory parameters were measured within ± four weeks of the vitamin D analysis. These parameters included CRP, white blood cells, granulocytes (neutrophils, eosinophils, and basophilic granulocytes), lymphocytes, and mononuclear cells. Internal quality controls of repeated vitamin D measurements in individual patients showed that the vitamin D concentrations in the individual subjects changed only minimally within 4 weeks. Therefore, we chose a time window of +/− 4 weeks around the blood-taking of the vitamin D sample for cohorts A and B (see below) to significantly increase the number of evaluable cases and thus improve the power of the study. In total, cohort A included 455,062 patients.

In the second cohort, cohort B, we examined the link between vitamin D and the parameters of calcium and phosphate metabolism. This part of the study included patients who received clinical laboratory parameters measured within a ± four-week window of the vitamin D analysis, namely total cholesterol, low-level lipoprotein, high-level lipoprotein, calcium, phosphate, iPTH, creatinine, and hemoglobin, totaling 47,778 patients. This further analysis could contribute to a broader understanding of vitamin D levels connected with metabolic health.

### 2.2. Clinical and Laboratory Parameters

Abbott Architect i2000 electrochemiluminescence immunoassay (Abbott Laboratories, Wiesbaden, Germany) was used to measure the levels of 25(OH)D. Similarly, the quantities of 1,25(OH)_2_D were determined using a chemiluminescent immunoassay with the Abbott Architect i2000. Calcium and phosphate concentrations were quantified through a photometric assay with the AU5800 clinical chemistry system (Beckman Coulter GmbH, Krefeld, Germany). The calcium-phosphate product (Ca × P) was determined by the formula: calcium (mmol/L) × 4 × Phosphate (mmol/L) × 3.1 = calcium × phosphate (mg^2^/dL^2^). The levels of iPTH were measured with an electrochemiluminescence immunoassay (ECLIA) (Roche, Wiesbaden, Germany), and CRP concentrations were determined using a turbidimetry assay. The WBC count was conducted using the Beckman Coulter UniCel DxH 800 analyzer (Beckman Coulter, Brea, CA, USA) for automated hematology analysis. Additional parameters were quantified employing standardized methodologies from IMD Berlin (https://www.imd-berlin.de/ accessed on 29 May 2024). Rigorous protocols for quality assurance and control were implemented across all clinical and laboratory data.

### 2.3. Statistical Analyses

SPSS version 23.0 (IBM, New York, NY, USA) was applied for the statistical analyses. Variables were reported as mean ± standard deviation (SD), except sex. The parameters were categorized based on deciles of serum 25(OH)D or 1,25(OH)_2_D concentrations, and average values of 25(OH)D or 1,25(OH)_2_D, and subsequently depicted in graphical format with a curve estimation analysis (linear, exponential, and quadratic models). The dependency of serum concentrations of vitamin D metabolites was analyzed by calculating the average concentrations of vitamin D metabolites for each month of the year based on our own data. Using these data and references to the relevant literature [29], we established two groups of months of the year (low average versus high average vitamin D metabolites groups). In addition, we performed multiple linear regression analyses with parameters in calcium-phosphorus metabolism (serum calcium/serum phosphate/iPTH) as well as in inflammatory response (CRP/WBC) as dependent variables. Age, gender, seasonal grouping, and two vitamin D concentrations (25(OH)D or 1,25(OH)_2_D) were incorporated into the regression model. To avoid omissions and to capture both linear as well as non-linear relationships between vitamin D, as well as the above parameters, we entered both the vitamin D concentration and the square root of the vitamin D concentration into the multiple linear regression model [30]. A significance criterion of *p* < 0.05 was established. A graphical representation of the correlation curves was created using data defined as mean ± standard error of the mean (SEM) and V8 of GraphPad Prism (GraphPad Software, Inc., San Diego, CA, USA).

### 2.4. Potential Bias

Although this study is one of the largest studies aiming to establish target ranges for 25(OH) vitamin D and also 1,25 (OH)_2_ vitamin D, the study had several potential biases that need to be considered:

The study population was not selected randomly, hence, although very large, it may not be representative of the general population. The data were collected from patients at specific medical institutions, which might not reflect the broader demographics or health conditions of the general population.

The study may not have adequately controlled for all potential confounders. Factors such as diet, genetic predispositions, and the use of supplements can influence vitamin D levels and health outcomes, potentially skewing the results.

The cross-sectional nature of the study limits the ability to draw causal inferences. The data represent a single point in time, making it difficult to determine the direction of the relationship between vitamin D levels and health outcomes.

The study was conducted in Germany, where seasonal variations in sunlight can significantly affect vitamin D levels. This geographical limitation may not make the findings applicable to populations in different regions with varying levels of sun exposure.

## 3. Results

Cohort A included 455,062 participants (64.4% females, 35.6% males) with an average age of 53.67 years. The average 125(OH)2D was 54.13 pg/mL, and 25(OH)D was 27.11 ng/mL. Average WBC was 6.87 Gpt/L, CRP 4.17 mg/L. Other markers were: granulocytes 0.03 Gpt/L, neutrophils 3.87 Gpt/L, eosinophils 0.18 Gpt/L, basophilic granulocytes 0.05 Gpt/L, lymphocytes 2.00 Gpt/L, and mononuclear cells 0.55 Gpt/L.

Cohort B had 47,778 participants (53.6% females, 46.4% males), with an average age of 59.00 years. The average 125(OH)2D was 51.60 pg/mL, and 25(OH)D was 29.60 ng/mL. Average calcium was 2.35 mmol/L, phosphate 1.09 mmol/L, and Ca × P 31.75 mg^2^/dL^2^. iPTH averaged 58.16 pg/mL. Additional parameters were: creatinine 1.18 mg/dL, HDL 55.23 mg/dL, LDL 116.42 mg/dL, total cholesterol 187.99 mg/dL, and hemoglobin 13.51 g/dL; for more details of both cohorts see Table 2 and Figure 1. All clinical laboratory parameters were within the healthy population’s reference range. Seasonal dependency of serum vitamin D metabolite concentrations are illustrated in Figure 2.

For the investigation of the relationship between inflammation and vitamin D, various curve-fitting models—linear, exponential, and quadratic—were employed. Among these, the quadratic model provided the highest R^2^ value, indicating the best fit (Table 3a). The relation between several inflammatory markers and 1,25(OH)_2_D showed that quadratic models consistently yielded *p*-values below 0.05, highlighting statistically significant correlations. Similarly, for 25(OH)D, significant relationships were found in all quadratic models except for granulocytes. There was a U-shaped (inverse quadratic) relationship between inflammatory biomarkers such as CRP (R^2^ = 0.795; *p* = 0.004) and WBC count (R^2^ = 0.883; *p* = 0.001) with 25(OH)D. With 25(OH)D values of 32 ng/mL and 42 ng/mL, respectively, there were minimal WBC and CRP counts.

The cohort was further divided into deciles based on the concentrations of 1,25(OH)_2_D and 25(OH)D, and the mean ± SEM for each indicator was plotted (Figure 3a). A multivariate regression analysis was conducted to investigate the effects of 25(OH)D and 1,25(OH)_2_D on inflammatory response parameters (WBC/CRP), incorporating age, gender, seasonal grouping, and both the original and square root-transformed vitamin D variables. According to Table 4a, both original and transformed vitamin D variables significantly influenced metabolic parameters independently of other factors. The unstandardized coefficients B for the square roots of vitamin D were larger than those for the unaltered vitamin D measurements, suggesting a more pronounced non-linear link between CRP/WBC and vitamin D levels, and indicating that the transformation makes the model more sensitive to subtle variations in vitamin D’s effects.

Additionally, the study analyzed the effect of vitamin D related to metabolism of phosphate and calcium using linear, exponential, and quadratic models. Consistent with the findings on inflammation, the quadratic model again demonstrated superior predictive power with the highest R^2^ values (Table 3b). All quadratic correlations between calcium-phosphate metabolism components and 25(OH)D were statistically worthwhile (*p* < 0.05). However, 1,25(OH)_2_D’s associations with total cholesterol and hemoglobin did not show significant linear relationships (correspondingly, *p* = 0.092 and *p* = 0.071).

Subdividing the cohort into deciles based on 10% increments of concentrations of 1,25(OH)_2_D and 25(OH)D provided a detailed graphical representation of the findings, displaying mean ± SEM (Figure 3b). Multiple regression analyses, which included age, sex, seasonal grouping, 1.25(OH)2D, 25(OH)D, and the square roots of these, with calcium and phosphorus metabolism parameters (serum calcium, serum phosphate, and iPTH) as dependent variables, showed that vitamin D has substantial, independent effects on these parameters (Table 4b). The larger unstandardized coefficients B for the square roots of vitamin D underscore a sensitive non-linear relationship between vitamin D and metabolic parameters.

## 4. Discussion

Target 25(OH)D concentrations are the subject of ongoing debate among physicians and scientists. Guidelines from international societies based largely on vitamin D-dependent processes (bone growth in children, bone fracture risk, healing of fractures, but also risk assessments of 25(OH)D’s effects without regard to bone health) provide good suggestions for lower and upper 25(OH)D threshold values (Table 1). We analyzed the topic with a slightly different approach. In two large data collections of clinical biochemical parameters, we assessed the link between target parameters and 1,25(OH)_2_D or 25(OH)D that describe inflammation (cohort A, 455,062 subjects) and mineral metabolism (cohort B, 47,778 subjects). The results are compatible with 30–50 ng/mL as the goal values for 25(OH)D if selecting tight target values, or 25–60 ng/mL using more flexible boundaries and 1,25(OH)_2_D target values of 50–70 pg/mL, or 40–80 pg/mL with wider boundaries. However, it should be emphasized that there is no exact formula to analyze or set the target values for both 1,25(OH)_2_D and 25(OH)D. The proposals from our study for target values are based on the integrated study of the links between the vitamin D system information and mineral metabolism. The target values determined for 25(OH)D fit well with mortality studies on serum 25(OH)D concentrations in Europe and the USA, and with data from people living in Africa under optimal sun exposure, where the human vitamin D system has developed over several hundred thousand years of evolution.

### 4.1. Vitamin D and Inflammation

Both the CRP and WBC count are markers of the body’s inflammatory and immunological responses. The way that vitamin D affects the immune system is what drives the link between vitamin D, CRP, and WBC count. The liver produces CRP, a protein, in reaction to inflammation. There is a reverse link between 25(OH)D and CRP levels, according to several pieces of research. Lower CRP levels have been linked to higher 25(OH)D levels, indicating that vitamin D may have anti-inflammatory properties [31,32]. White blood cells are essential immune system components involved in fighting infections and responding to inflammation. 1,25(OH)_2_D and 25(OH)D have been demonstrated to modulate the function of various types of white blood cell, including lymphocytes and macrophages. Research has indicated that immunological dysregulation might result from a 25(OH)D deficit, resulting in an increased WBC count as the body attempts to compensate for the deficiency and combat inflammation or infection [33,34]. However, sufficient levels of 25(OH)D correspond to proper immune function and adequate regulation of the WBC count [35].

Several recent studies have reported an inverse link between 1,25(OH)_2_D and 25(OH)D, and either CRP or WBC [13,14], however, other studies could not confirm this association [15,16]. Our study’s strength is due to the size of the study population, allowing for a powerful analysis. Our study is the first to demonstrate the existence of an inverted quadratic (U-shaped) connection between inflammatory biomarkers such as CRP (*p* = 0.004; R^2^ = 0.795) and WBC (R^2^ = 0.883; *p* = 0.001) with 25(OH)D. At 25(OH)D values between 32 ng/mL and 42 ng/mL, very little CRP and WBC were seen. The minimal WBC count at 42ng/mL 25(OH)D corresponded to a white blood cell concentration of 6.7 Gp/L. Remarkably, this concentration of white blood cells has been related to optimal survival rates/all-cause death in research with healthy people, but also people with heart failure. In a UK biobank study, 425,264 individuals were followed up for all-cause mortality. A threshold of 6.1 Gp/L was that at which minimal hazard ratios for all-cause death were seen in relation to white blood cell counts; higher all-cause mortality was linked to both lower and higher WBC levels [36]. A TOPCAT (Treatment of Heart Failure with Preserved Cardiac Function Using an Aldosterone Antagonist) experiment involving 2898 heart failure patients came to virtually identical results regarding the optimal survival rate in relation to the white blood cell count [37]. These clinical studies, together with our data, are consistent with target 25(OH)D levels ranging from 30 to 50 ng/mL. In the JUPITER trial, a large, randomized study of males who appeared to be in good health, and women without hyperlipidemia, found that participants who received rosuvastatin compared to those who received a placebo had significantly lower odds of a first major cardiovascular event and all-cause death, although with raised levels of high-sensitivity CRP (hsCRP). This remarkable clinical effect was linked to a significant reduction of hsCRP from between 4.2 and 1.8 mg/L [38]. The study also highlights the impact of relatively small changes in CRP—comparable with the relationships found in our study between CRP and 25(OH)D.

Regarding 1,25(OH)_2_D, CRP (R^2^ = 0.915; *p* < 0.001) and WBC (R^2^ = 0.943; *p* < 0.001) showed a comparable U-shaped connection. At a 1,25(OH)_2_D concentration of 60 pg/mL, both CRP and WBC were at minimal levels (Figure 3a). Both too high and more CRP and WBCs were connected with too low 25(OH)D and 1,25(OH)_2_D concentrations. Impactful studies analyzing overall mortality from all causes in the broader population with respect to the level of 1,25(OH)_2_D are missing so far.

In addition, 1,25(OH)_2_D and 25(OH)D impact critical inflammation-associated signaling pathways, including NF-κB, that regulate inflammatory signaling, thus preventing the escalation of inflammatory responses [39]. Interesting and fitting to the concept of a link in the U-shaped form between inflammation and 25(OH)D/1,25(OH)_2_D is a new basic science study that suggests a similar U-shaped link between blood pressure, renal function, and morphology, with the NF-κB pathway’s activation utilizing mice models with NF-κB pathways that are partly active and inactivated [40]. These studies suggest an optimal target range and that both too high and too low levels of certain parameters could be associated with negative outcomes.

### 4.2. Calcium-Phosphate Metabolism

Concerning cohort B, it is important to highlight that elevated calcium-phosphate product (Ca × P) levels are a risk factor associated with vascular calcification and mortality. This holds for the general population and even more so among individuals suffering from chronic kidney disease [41,42,43]. Increased blood calcium and phosphate concentrations can cause calcium phosphate crystals to form in blood vessels, a process known as vascular calcification. This condition is linked to a higher risk of cardiovascular disease and death. Hence, it is crucial to regulate the concentration of the Ca × P, particularly in individuals suffering from chronic kidney disease and related disorders. Our findings, as illustrated in Figure 3b, indicate that while Ca and P values both remained within the reference range, there was a notable increase in Ca × P corresponding to elevated levels of 25(OH)D. This pattern became most noticeable when the 25(OH)D level rose above the 50 ng/mL cutoff, which caused the Ca × P to spike. Given the adverse impacts associated with elevated Ca × P levels, particularly when chronic renal disease is present, our recommendation is to maintain vitamin D concentrations within a more cautious range under 50 ng/mL.

Vitamin D also exerts regulatory effects on iPTH levels, primarily by inhibiting its synthesis in the parathyroid glands. iPTH levels are therefore a biomarker of a sufficient level of vitamin D in patients without a primary parathyroid disease. In our data, we saw a reverse relationship between PTH and 1,25(OH)_2_D and 25(OH)D, which is in line with earlier research [44,45,46]. According to the providers of the iPTH assay used in this study, the normal range for iPTH is 15–65 pg/mL, and according to our study’s findings, 1,25(OH)_2_D and 25(OH)D levels greater than 30 pg/mL and more than 10 ng/mL, respectively, are consistent with normal iPTH levels (Figure 3b).

### 4.3. Out of Africa Hypothesis of Humans and Vitamin D

Regarding the ideal level of 25(OH)D, published data suggest that the average concentration of 25(OH)D in non-human primates and humans living in Africa with full exposure to sunshine is around 30–60 ng/mL [47,48,49,50,51], which corresponds to the target range from our present findings. It is well known that humans originated from Africa and evolved from primate ancestors, and that modern humans migrated to Europe and Asia about 45,000 years ago. Over several hundred thousand years of human evolution, vitamin D metabolism-related metabolic genes were optimized under evolutionary pressure, resulting in optimal levels of 25(OH)D of between 30 and 60 ng/mL for healthy survival, and the same has occurred in non-human primates. In other words, the vitamin D system optimized by human evolution over many generations has produced the selection of optimal 25(OH)D serum concentrations for healthy survival. These concentrations correspond well to our suggested target of 25(OH)D concentrations by means of an investigation of the biological results of 25(OH)D against biochemical parameters.

### 4.4. Mortality and Vitamin D Levels

International guidelines make statements about upper and lower limits for vitamin D, which should not be exceeded or undercut, but they do not give clear guidance on ideal levels of 25(OH)D. One way to address this question is to analyze which serum 25(OH)D concentration is linked to the lowest cardiovascular and all-cause mortality. Several cross-sectional epidemiological studies showed a fairly consistent picture: in a general US population study with 15,099 individuals aged 20 and above, and 3784 death cases following a follow-up of 15 years, a reverse J-shaped relationship between serum 25(OH)D levels and all-cause death was discovered [52,53]. A comprehensive review of European research showed comparable results to the US study [54]. The minimal mortality in these studies was linked to 30–40 ng/mL 25(OH)D concentrations, which corresponds with our findings.

### 4.5. Study Strengths and Limitations

The scale of the study is one of its main advantages, particularly the size of cohort A, clearly showing a U-shaped correlation between inflammatory indicators and forms of vitamin D, CRP, and WBC.

Several disadvantages should also be acknowledged. Our study has a cross-sectional nature without follow-up data, as the data used in our study are from our clinical laboratory data storage system. Given that, we do not have pure clinical data on underlying diseases such as vitamin D supplementation or calcium intake, or data on pre-existing diseases such as diabetes or hypertension. It is likewise a disadvantage of the study that we did not have data on vitamin K2 uptake and/or serum concentrations, since vitamin K2 is also involved in vascular calcification/mineral–bone metabolism [55]. Our available data only allowed the consideration of creatinine, calcium, phosphate, and PTH as confounding factors in the statical models describing this endocrine network. Our methodology also has the disadvantage that there is no exact formula to calculate or set the target values about 1,25(OH)_2_D and 25(OH)D. Suggested targets are based on an integrative view of the individual correlation analysis with a much stronger consideration of cohort A simply due to its size. The proposals from our study for target values are based on the integrated analysis of the links between mineral metabolism, information, and the vitamin D system. This investigation was carried out within the general German population; findings may vary when applied to specific subgroups, different racial groups, or distinct disease states. Finally, we used data on overall 25(OH)D levels, whereas the analysis of unrestricted 25(OH)D (which is the bioactive form of 25(OH)D entering cells and communicating with the nuclear vitamin D receptor) could also have significant importance [56].

### 4.6. Target Range for 1,25(OH)_2_D and 25(OH)D

Based mainly on the U-shaped relationship observed in cohort A between inflammatory biomarkers and 25(OH)D like WBC and CRP counts in our study, we suggest between 30 and 50 ng/mL of serum 25(OH)D as tight target concentrations for 25(OH)D or 25–60 ng/mL, making the boundaries a little more flexible in the general population. This suggestion fits the existing literature from other investigations [23,57]. Our data on calcium phosphate metabolism are also consistent with this suggestion. However, the power of the calcium-phosphate sub-study was substantially lower because the first cohort focusing on 455,062 participants made up the 25(OH)D/1,25(OH)_2_D and inflammatory groups, whereas the second cohort focusing on 47,778 participants participated in the mineral metabolism and 25(OH)D/1,25(OH)_2_D studies.

We found that 1,25(OH)_2_D and the inflammatory markers CRP and WBC had a U-shaped connection, just like 25(OH)D. The minimal levels of CRP and WBC count were observed at around 60 pg/mL of 1,25(OH)_2_D. Considering the quadratic relationship, the target range for 1,25(OH)_2_D should aim to be maintained around this concentration. The current literature suggests that optimal levels of 1,25(OH)_2_D are essential for bone health and calcium balance [58]. However, specific target ranges for 1,25(OH)_2_D are less well-defined compared to 25(OH)D.

## 5. Conclusions

Based mostly on our cohort A, which relied on 25(OH)D and inflammation biomarkers, but also consistent with the data from the mineral metabolism group, we suggest target values ranging from 30 to 50 ng/mL for 25(OH)D if selecting tight target values, or 25–60 ng/mL using more flexible boundaries, also taking into account the existing literature on survival data regarding 25(OH)D and 1,25(OH)_2_D concentrations seen in people living in Africa under optimal sun exposure, where the human vitamin D system developed. From our study, a suggestion for target 1,25(OH)_2_D values could be 50–70 pg/mL or 40–80 pg/mL, with wider boundaries, but this needs more clinical studies to be confirmed.

## Figures and Tables

**Figure 1 nutrients-16-02679-f001:**
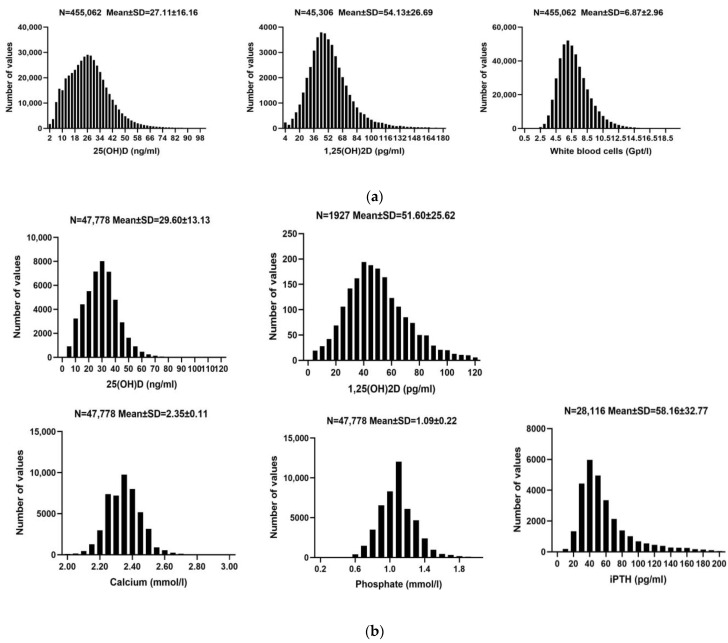
Distribution of 25 (OH)D, 1,25(OH)_2_D concentrations, and other measured parameters. (**a**) Cohort A. Distribution of 25(OH)D and 1,25(OH)_2_D concentrations, and white blood cell count in the cohort related to inflammation. Abbreviations: 25(OH)D: 25-hydroxy-vitamin D; 1,25(OH)_2_D: 1,25-dihydroxy-vitamin D. (**b**) Cohort B. Distribution of 25(OH)D, 1,25(OH)_2_D, calcium, phosphate, and intact parathyroid hormone (iPTH) concentrations in the cohort related to calcium and phosphate metabolism. Abbreviations: 25(OH)D: 25-hydroxy-vitamin D; 1,25(OH)_2_D: 1,25-dihydroxy-vitamin D;iPTH: intact parathyroid hormone.

**Figure 2 nutrients-16-02679-f002:**
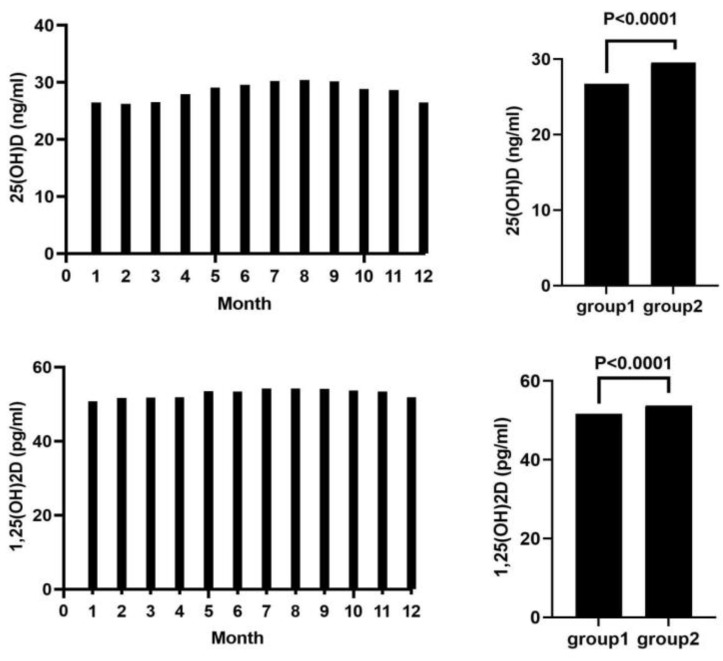
Seasonal analysis of vitamin D. Left side: mean monthly concentration of 25(OH)D (**top**) and 1,25(OH)_2_D (**bottom**). Group 1, vitamin D metabolite concentrations from December to April; Group 2, vitamin D metabolite concentrations from May to November. Comparing the 25(OH)D and 1,25(OH)_2_D concentrations between these two groups revealed higher concentrations in the summer season for both 25(OH)D and 1,25(OH)_2_D. Abbreviations: 25(OH)D: 25-hydroxy-vitamin D; 1,25(OH)_2_D: 1,25-dihydroxy-vitamin D.

**Figure 3 nutrients-16-02679-f003:**
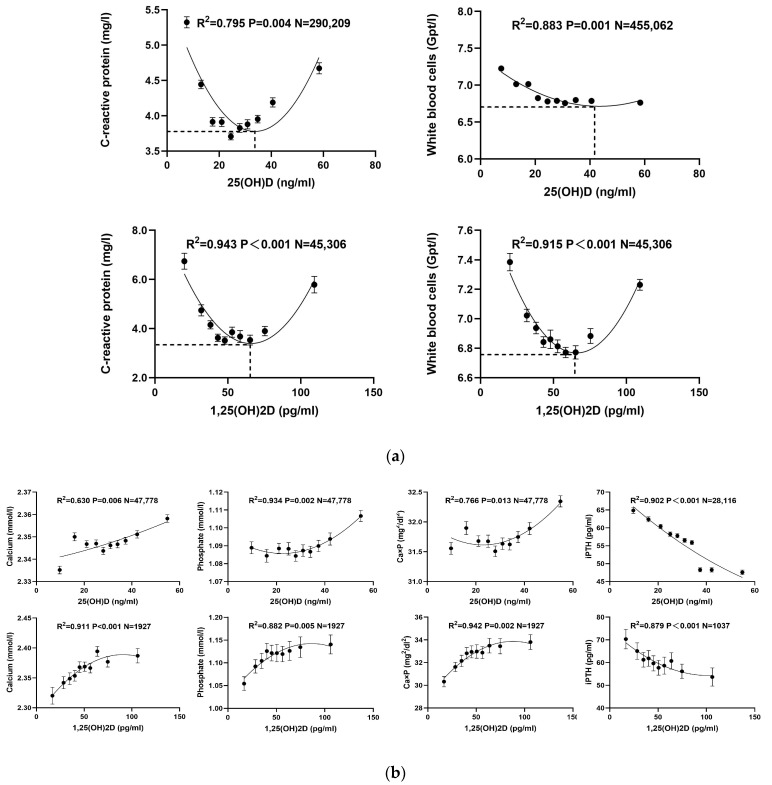
Association between vitamin D forms and various parameters. (**a**) Cohort A. Curve fitting analysis of the relationship between 25(OH)D/1,25(OH)_2_D concentrations and each inflammatory parameter using linear, exponential, and quadratic correlation models. Quadratic models fit the best (see Section 3). There was a U-shaped (inverse quadratic) relationship between inflammatory biomarkers such as CRP (R^2^ = 0.795; *p* = 0.004) and WBC count (R^2^ = 0.883; *p* = 0.001) with 25(OH)D. Minimal CRP and WBC counts were seen at 25(OH)D concentrations of 32 ng/mL and 42 ng/mL, respectively. Abbreviations: 25(OH)D: 25-hydroxy-vitamin D; 1,25(OH)_2_D: 1,25-dihydroxy-vitamin D. (**b**) Cohort B. Curve fitting analysis of the relationship between 25(OH)D/1,25(OH)_2_D concentrations and each parameter of the calcium/phosphate metabolism through linear, exponential, and quadratic correlation models. Quadratic models fit the best (see Section 3). iPTH correlated inversely with 25(OH)D (R^2^ = 0.902; *p* < 0.001) and 1,25(OH)_2_D (R^2^ = 0.879; *p* < 0.001), whereas calcium correlated positively with 25(OH)D (R^2^ = 0.630; *p* = 0.006) and 1,25(OH)_2_D (R^2^ = 0.911; *p* < 0.001). Phosphate likewise correlated positively with 25(OH)D (R^2^ = 0.934; *p* = 0.002) and 25(OH)_2_D (R^2^ = 0.882; *p* = 0.005). Abbreviations: 25(OH)D: 25-hydroxy-vitamin D; 1,25(OH)_2_D: 1,25-dihydroxy-vitamin D; iPTH: intact parathyroid hormone; Ca × P: calcium-phosphate product.

**Table 1 nutrients-16-02679-t001:** Vitamin D guidelines published in the last 20 years.

	Year	Target Disease	Concentration (ng/mL)	Explanation
Holick [19]	2004	General health care	30–50	optimal range
			≥150	toxicity
Holick [20]	2005	Bone health	<32	deficiency
			≥150	toxicity
Wang et al. [21]	2008	Cardiovascular Disease	<10	significant risk
			>30	relative safety
Norman et al. [22]	2010	General health care	<5	severe deficiency
			5–10	deficiency
			10–30	insufficiency
			30–100	optimal range
			100–150	probable toxicity
			≥300	apparent toxicity
Ross et al. [23]	2010	General health care	20–50	optimal range
Pérez-López et al. [24]	2012	General health care	<20	deficiency
			<30	insufficiency
Rizzoli et al. [25]	2013	General health care	<10	deficiency
			<20	insufficiency
Cosman et al. [26]	2014	Osteoporosis	<30	insufficiency
Munns et al. [27]	2016	General health care	<12	deficiency
			<20	insufficiency
			≥100	toxicity
Giustina et al. [28]	2019	Bone health	<12	deficiency
			<20	insufficiency

Important studies on 25(OH)D concentrations over the past 20 years.

**Table 2 nutrients-16-02679-t002:** Baseline patient characteristics. (a) Cohort A. (b) Cohort B.

**(a) Cohort A**			
	**N (N = 455,062)**	**Mean ± SD**	**Reference Range**
Sex	Males: 162,019 (35.6%)	-	
	Females: 293,043 (64.4%)		-
Age (years)	455,062	53.67 ± 19.80	-
25(OH)D (ng/mL)	455,062	27.11 ± 16.16	30.00–100.00
1,25(OH)_2_D (pg/mL)	45,306	54.13 ± 26.69	19.90–79.30
WBC (Gpt/L)	455,062	6.87 ± 2.96	3.6–28.2
CRP (mg/L)	290,209	4.17 ± 10.83	<5
Granulocytes (Gpt/L)	198,928	0.03 ± 0.01	-
Neutrophils (Gpt/L)	205,121	3.87 ± 1.65	1.3–22.3
Eosinophils (Gpt/L)	203,224	0.18 ± 0.06	0.02–1.10
Basophilic granulocytes (Gpt/L)	204,587	0.05 ± 0.03	<0.35
Lymphocytes (Gpt/L)	205,120	2.00 ± 0.75	1.1–13.6
Mononuclear cells (Gpt/L)	205,120	0.55 ± 0.19	0.10–2.70
**(b) Cohort B**			
	**N (N = 47,778)**	**Mean ± SD**	**Reference Range**
Sex	Males: 22,161 (46.4%)	-	
	Females: 25,617 (53.6%)		-
Age (years)	47,778	59.00 ± 19.51	-
25(OH)D (ng/mL)	47,778	29.60 ± 13.13	30.00–100.00
1,25(OH)_2_D (pg/mL)	1927	51.60 ± 25.62	19.90–79.30
Calcium (mmol/L)	47,778	2.35 ± 0.11	1.90–2.75
Phosphate (mmol/L)	47,778	1.09 ± 0.22	0.81–2.42
Ca × P (mg^2^/dL^2^)	47,778	31.75 ± 6.72	-
iPTH (pg/mL)	28,116	58.16 ± 32.77	15.00–65.00
Creatinine (mg/dL)	40,496	0.88 ± 0.42	0.16–1.95
LDL (mg/dL)	32,074	114.42 ± 40.20	<115.00
HDL (mg/dL)	31,311	55.23 ± 17.73	>45.00
TC (mg/dL)	30,844	187.99 ± 45.52	<200.00

Cohort A was focused on the relationship between vitamin and inflammatory parameters. Abbreviations: 25(OH)D: 25-hydroxy vitamin D; 1,25(OH)_2_D: 1,25-dihydroxy vitamin D; WBC: white blood cell; CRP: C-reactive protein. The reference range is from the Berlin Institute for Medical Diagnostics (https://www.imd-berlin.de). Cohort B was focused on the relationship between vitamin D and the calcium and phosphate metabolism. Abbreviations: 25(OH)D: 25-hydroxy-vitamin D; 1,25(OH)_2_D: 1,25-dihydroxy-vitamin D; SD: standard deviation; iPTH: intact parathyroid hormone; Ca × P: calcium-phosphate product; LDL: low-density lipoprotein; HDL: high-density lipoprotein; TC: total cholesterol. The reference range is from the Berlin Institute for Medical Diagnostics (https://www.imd-berlin.de).

**Table 3 nutrients-16-02679-t003:** Curve estimation of serum 25(OH)D and 1,25(OH)_2_D. (a) Cohort A. (b) Cohort B.

**(a) Cohort A**						
**25(OH)D**						
**Parameters**	**Linear**		**Exponential**		**Quadratic**	
	**R^2^**	** *p* **	**R^2^**	** *p* **	**R^2^**	** *p* **
WBC (Gpt/L)	0.512	0.020	0.523	0.019	0.883	0.001
CRP (mg/L)	0.021	0.731	0.012	0.785	0.795	0.004
Granulocytes (Gpt/L)	0.177	0.236	0.165	0.244	0.515	0.079
Neutrophils (Gpt/L)	0.243	0.148	0.242	0.150	0.658	0.024
Eosinophils (Gpt/L)	0.863	<0.001	0.878	<0.001	0.992	<0.001
Basophilic granulocytes (Gpt/L)	0.549	0.014	0.553	0.014	0.852	0.001
Lymphocytes (Gpt/L)	0.934	<0.001	0.943	<0.001	0.995	<0.001
Mononuclear cells (Gpt/L)	0.652	0.005	0.652	0.005	0.827	0.002
**1,25(OH)_2_D**						
WBC (Gpt/L)	0.004	0.861	0.004	0.865	0.943	<0.001
CRP (mg/L)	0.004	0.864	0.002	0.906	0.915	<0.001
Granulocytes (Gpt/L)	0.006	0.828	0.004	0.869	0.991	<0.001
Neutrophils (Gpt/L)	0.031	0.627	0.028	0.641	0.971	<0.001
Eosinophils (Gpt/L)	0.927	<0.001	0.944	<0.001	0.948	<0.001
Basophilic granulocytes (Gpt/L)	0.870	<0.001	0.888	<0.001	0.970	<0.001
Lymphocytes (Gpt/L)	0.065	0.476	0.068	0.469	0.619	0.034
Mononuclear cells (Gpt/L)	0.282	0.116	0.277	0.119	0.970	<0.001
**(b) Cohort B**						
**25(OH)D**						
**Parameters**	**Linear**		**Exponential**		**Quadratic**	
	**R^2^**	** *p* **	**R^2^**	** *p* **	**R^2^**	** *p* **
Calcium (mmol/L)	0.624	0.007	0.604	0.019	0.630	0.006
Phosphate (mmol/L)	0.565	0.012	0.646	0.010	0.934	0.002
Ca × P (mg^2^/dL^2^)	0.447	0.035	0.602	0.022	0.766	0.013
iPTH (pg/mL)	0.892	<0.001	0.883	<0.001	0.902	<0.001
Creatinine (mg/dL)	0.983	<0.001	0.980	<0.001	0.983	<0.001
LDL (mg/dL)	0.795	<0.001	0.794	<0.001	0.795	<0.001
HDL (mg/dL)	0.923	<0.001	0.953	<0.001	0.965	<0.001
TC (mg/dL)	0.785	<0.001	0.794	<0.001	0.806	<0.001
Hemoglobin (g/dL)	0.787	<0.001	0.817	<0.001	0.835	<0.001
**1,25(OH)_2_D**						
Calcium (mmol/L)	0.721	0.002	0.876	0.001	0.911	<0.001
Phosphate (mmol/L)	0.648	0.005	0.723	0.005	0.882	0.005
Ca × P (mg^2^/dL^2^)	0.718	0.002	0.827	0.002	0.942	0.002
iPTH (pg/mL)	0.774	<0.001	0.854	<0.001	0.879	<0.001
Creatinine(mg/dL)	0.614	0.007	0.856	0.005	0.977	0.003
LDL (mg/dL)	0.200	0.195	0.696	0.010	0.727	0.007
HDL (mg/dL)	0.720	0.002	0.803	0.002	0.835	0.001
TC (mg/dL)	0.264	0.128	0.333	0.104	0.473	0.092
Hemoglobin(g/dL)	0.157	0.257	0.344	0.131	0.654	0.071

Cohort A was focused on the relationship between vitamin and inflammatory parameters. Abbreviations: 25(OH)D: 25-hydroxy-vitamin D; 1,25(OH)_2_D: 1,25-dihydroxy-vitamin D; WBC: white blood cell; CRP: C-reactive protein. Cohort B was focused on the relationship between vitamin D and the calcium and phosphate metabolism. Abbreviations: 25(OH)D: 25-hydroxy-vitamin D; 1,25(OH)_2_D: 1,25-dihydroxy-vitamin D; iPTH: intact parathyroid hormone; Ca × P: calcium-phosphate product; LDL: low-density lipoprotein; HDL: high-density lipoprotein; TC: total cholesterol.

**Table 4 nutrients-16-02679-t004:** (a) Multivariate regression for cohort A. (b) Multivariate regression for cohort B.

**(a)**												
	**WBC**						**CRP**					
	** *p* **		**B ^#^**		**95%CI**		** *p* **		**B ^#^**		**95%CI**	
Constant	0.000		9.029		8.705~9.353		<0.001		10.162		8.643~11.681	
Gender	0.009		−0.082		−0.144~−0.020		0.031		−0.324		−0.619~−0.029	
Age (years)	0.364		0.001		−0.001~0.002		<0.001		0.046		0.038~0.053	
Seasonal group	0.900		0.004		−0.054~0.062		0.042		0.293		0.011~0.576	
25(OH)D (ng/mL)	0.033		0.004		0.001~0.007		<0.001		0.038		0.018~0.059	
Sqrt-25(OH)D	<0.001		−0.157		−0.210~−0.104		<0.001		−0.716		−0.977~−0.456	
1,25(OH)_2_D (pg/mL)	<0.001		0.027		0.022~0.031		<0.001		0.104		0.083~0.124	
Sqrt-1,25(OH)_2_D	<0.001		−0.383		−0.453~−0.313		<0.001		−1.557		−1.886~−1.229	
**(b)**												
	**Calcium**				**Phosphate**				**iPTH**			
	** *p* **	**B** ** ^#^ **		**95%CI**	** *p* **	**B** ** ^#^ **		**95%CI**	** *p* **	**B** ** ^#^ **		**95%CI**
Constant	0.000	2.307		2.247~2.367	<0.001	1.432		1.330~1.535	<0.001	210.799		161.571~260.027
Gender	0.005	0.020		0.006~0.034	<0.001	0.108		0.085~0.132	<0.001	−24.417		−35.729~−13.105
Age (years)	<0.001	−0.001		−0.001~0.000	<0.001	−0.005		−0.005~−0.004	0.710	0.057		−0.246~0.360
Seasonal group	0.941	−0.001		−0.014~0.013	0.074	−0.021		−0.044~0.002	0.307	−5.673		−16.560~5.215
25(OH)D (ng/mL)	0.004	0.001		0.000~0.002	<0.001	−0.002		−0.003~−0.001	0.007	−0.558		−0.961~−0.155
Sqrt-25(OH)D	0.008	0.010		0.003~0.017	<0.001	0.054		0.037~0.070	<0.001	13.071		5.446~20.696
1,25(OH)2D (pg/mL)	0.045	0.002		0.001~0.003	<0.001	0.004		0.003~0.005	<0.001	0.965		0.550~1.379
Sqrt-1,25(OH)2D	0.012	0.014		0.003~0.025	<0.001	−0.083		−0.102~−0.065	<0.001	−27.254		−35.016~−19.493

In cohort A we analyzed the relationship between vitamin D and inflammatory parameters. Table 2a shows that WBC and CRP are best fitted with quadratic curves in relation to 25(OH)D and 1,25(OH)_2_D, modeled by the equation y = ax^2^ + bx + c. To capture the non-linear relationship between vitamin D and inflammatory response parameters, both 25(OH)D and 1,25(OH)_2_D were square root-transformed and included in the analysis. The final model was adjusted for sex, age, seasonal group, 25(OH)D, Sqrt-25(OH)D, 1,25(OH)_2_D, and Sqrt-1,25(OH)_2_D. Seasonal group: the data were divided into two groups according to the level of vitamin D concentrations detected during the different seasons: the first group had low vitamin D concentrations from January to April, and in December; the second group had high vitamin D concentrations from May to November. Abbreviations: 25(OH)D: 25-hydroxy-vitamin D; 1,25(OH)_2_D: 1,25-dihydroxy-vitamin D; Sqrt-: square root-parameters; WBC: white blood cell; CRP: C-reactive protein; B ^#^: unstandardized coefficient B; 95%CI: confidence interval. In cohort B we analyzed the relationship between vitamin D and the calcium and phosphate metabolism. Table 2b shows that calcium/phosphate/iPTH are best fitted with quadratic curves in relation to 25(OH)D and 1,25(OH)_2_D, modeled by the equation y = ax^2^ + bx + c. To capture the non-linear relationship between vitamin D and calcium and phosphate metabolism parameters, both 25(OH)D and 1,25(OH)_2_D were square root-transformed and included in the analysis. The final model was adjusted for sex, age, seasonal group, 25(OH)D, Sqrt-25(OH)D, 1,25(OH)_2_D, and Sqrt-1,25(OH)_2_D. Seasonal group: the data were divided into two groups according to the level of vitamin D concentrations detected during the different seasons: the first group had low vitamin D concentrations from January to April, and in December; the second group had high vitamin D concentrations from May to November. Abbreviations: 25(OH)D: 25-hydroxy-vitamin D; 1,25(OH)_2_D: 1,25-dihydroxy-vitamin D; Sqrt-: square root-parameters; iPTH: intact parathyroid hormone; B ^#^: unstandardized coefficient B; 95%CI: confidence interval.

## Data Availability

All data generated or analyzed during this study are included in this article. Further enquiries can be directed to the corresponding author.

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
