# Peer review of "Target Values for 25-Hydroxy and 1,25-Dihydroxy Vitamin D Based on Their Associations with Inflammation and Calcium-Phosphate Metabolism"

_nutrients, 2024, doi:10.3390/nu16162679_

Round 1
Reviewer 1 Report
Comments and Suggestions for Authors
Thank you very much to the Editor of Nutrients for allowing me to review the paper entitled ‘Target Values for 25-Hydroxy and 1,25-Dihydroxy Vitamin D Based on Their Associations with Inflammation and Calcium-Phosphate Metabolism’.
The authors suggested target values for 25(OH)D between 25- 60 ng/ml and for 1,25(OH)2D values - between 40 – 80 pg/ml.
Major comments: The topic is important. This study is relatively well-designed. The statistical analyses seem correct but studies are based on relatively old literature (33 items out of 58 items were published more than ten years ago).
Results were given clearly with sufficient tables and data analysis, but under Figures and Tables, the abbreviations that were used should be explained.
A paragraph with limitations is missing (for example at the end of the discussion).
Access to most of the test results were not exactly at the same time but within a fairly large 4 week interval from the vitamin D level analyses.
The paper also does not address the role of vitamin K2 in bone, calcium and the role of vitamin D, despite a growing body of research indicating its crucial importance.
Lack of information about important role of vitamin K2 as the cofactor of γ-carboxylase (involve in γ-carboxylation of glutamine (Glu) residues in ostocalcin, resulting in γ-carboxyglutamic acid (Gla) residues that have an affinity for calcium ions, see inter alia Nikita Jadhav et al. Front Pharmacol. 2022; 13: 896920. doi: 10.3389/fphar.2022.896920).
The paper also fails to address the importance of vitamin D and calcium supplementation without adequate vitamin K2 level, in generating vascular calcification. Vitamin K2 may help reduce vascular calcification and thereby reduce the risk of cardiovascular disease [Schlieper G., et al. (2016). Vascular Calcification in Chronic Kidney Disease: an Update. Nephrol. Dial. Transpl. 31, 31–39. 10.1093/ndt/gfv111]
Minor comments
Instead of:
Africa under optimal sun exposure, where the human vitamin D system has developed.
Should be:
Africa is under optimal sun exposure, where the human vitamin D system has developed.
Instead of:
Ginde, A.A.; Mansbach, J.M.; Camargo, C.A. Association between serum 25-hydroxyvitamin D level and upper respiratory tract
infection in the Third National Health and Nutrition Examination Survey. Archives of internal medicine 2009, 169, 384-390.
Should be:
Ginde, A.A.; Mansbach, J.M.; Camargo, C.A. Association between serum 25-hydroxyvitamin D level and upper respiratory tract
infection in the Third National Health and Nutrition Examination Survey. Archives of Internal Medicine 2009, 169, 384-390.
Instead of:
14. de Oliveira, C.; Biddulph, J.P.; Hirani, V.; Schneider, I.J.C. Vitamin D and inflammatory markers: cross-sectional analyses using data from the English Longitudinal Study of Ageing (ELSA). Journal of nutritional science 2017, 6, e1.
Should be:
14. de Oliveira, C.; Biddulph, J.P.; Hirani, V.; Schneider, I.J.C. Vitamin D and inflammatory markers: cross-sectional analyses using data from the English Longitudinal Study of Ageing (ELSA). Journal of Nutritional Science 2017, 6, e1.
Instead of:
Michos, E.D.; Streeten, E.A.; Ryan, K.A.; Rampersaud, E.; Peyser, P.A.; Bielak, L.F.; Shuldiner, A.R.; Mitchell, B.D.; Post, W.
Serum 25-hydroxyvitamin d levels are not associated with subclinical vascular disease or C-reactive protein in the old order amish. Calcified tissue international 2009, 84, 195-202.
Should be:
Michos, E.D.; Streeten, E.A.; Ryan, K.A.; Rampersaud, E.; Peyser, P.A.; Bielak, L.F.; Shuldiner, A.R.; Mitchell, B.D.; Post, W. Serum 25-hydroxyvitamin D levels are not associated with subclinical vascular disease or C-reactive protein in the old order amish. Calcified Tissue International 2009, 84, 195-202.
Comments on the Quality of English LanguageMinor editing of English language required.
Reviewer 2 Report
Comments and Suggestions for Authors
The manuscript is interesting since the authors address and important theme in the nutritional field and endocrinology/metabolism. I have some minor observations/questions that might help improve the manuscript.
- How were participants selected, and what were the exclusion criteria?
- The statistical methods are robust, but additional information related to potential confounders is required. For example, were seasonal variations in vitamin D levels considered?
- Among the limitations of the study, I would add that the generalizability of the findings to populations outside Germany or to subgroups within the study population.
- line 86: why did you choose as a cut-off a 4 week maximal interval for collecting the datum of the inflammatory markers?
Reviewer 3 Report
Comments and Suggestions for Authors
The manuscript entitled ”Target Values for 25-Hydroxy and 1,25-Dihydroxy Vitamin D Based on Their Associations with Inflammation and Calcium-Phosphate Metabolism” describes a retrospective cohort study. Given the fact that there is a large number of subjects included, the results can emphasize epidemiological data on the studied topics.
However, some aspects need to be addressed before the manuscript is considered for publication:
a. Abstract – it should be structured and organized into subsections – Background and Objectives, Methods, Results, Conclusion
b. Introduction – while it presents sufficient background to motivate the scope of this research, the aims of this study should be clearly presented - e.g – “to establish target values for vitamin D metabolites”, “to understand the association between vitamin D levels and inflammation”, etc.
c. The Methods section should be logical structured, as follows: 1 – study design and setting (describe the study type, provide details about where the data was collected), 2 - study population (cohort definitions, criteria for inclusion, demographics, etc), 3 – data collection – biomarker measurements, equipment( calibration, etc.), 4 – statistical analysis, 5 – quality control – data integrity, how was the missing data handled,etc., 5 – ethical consideration – approval, consent, etc, 6 – address potential biases
d. Conclusion section should be inserted and emphasize the main outcomes described in the Discussion section
Round 2
Reviewer 3 Report
Comments and Suggestions for Authors
After reading the modifications we agreed that the authors have responded to our 4 comments. The missing info in the revision was in the Method section about the potential bias.
The rest of the topic being addressed we think that except for this issue, the rest of the articol is well documented.
Author Response
Comments:
After reading the modifications we agreed that the authors have responded to our 4 comments. The missing info in the revision was in the Method section about the potential bias.The rest of the topic being addressed we think that except for this issue, the rest of the articol is well documented.
Response:
Thank you for your comments, which we found very useful and correct, so we have added a description of the potential bias at the end of the "Methods" section of the article, highlighted in yellow (lines 147-163), and we have also re-uploaded our manuscript, please check it out.